# BricksRL: A Platform for Democratizing Robotics and Reinforcement Learning Research and Education with LEGO

**Sebastian Dittert**
Universitat Pompeu Fabra
sebastian.dittert@upf.edu

**Vincent Moens**
PyTorch Team, Meta
vincentmoens@gmail.com

**Gianni De Fabritiis**
ICREA, Universitat Pompeu Fabra
g.defabritiis@gmail.com

## Abstract

We present BricksRL, a platform designed to democratize access to robotics for reinforcement learning research and education. BricksRL facilitates the creation, design, and training of custom LEGO robots in the real world by interfacing them with the TorchRL library for reinforcement learning agents. The integration of TorchRL with the LEGO hubs, via Bluetooth bidirectional communication, enables state-of-the-art reinforcement learning training on GPUs for a wide variety of LEGO builds. This offers a flexible and cost-efficient approach for scaling and also provides a robust infrastructure for robot-environment-algorithm communication. We present various experiments across tasks and robot configurations, providing built plans and training results. Furthermore, we demonstrate that inexpensive LEGO robots can be trained end-to-end in the real world to achieve simple tasks, with training times typically under 120 minutes on a normal laptop. Moreover, we show how users can extend the capabilities, exemplified by the successful integration of non-LEGO sensors. By enhancing accessibility to both robotics and reinforcement learning, BricksRL establishes a strong foundation for democratized robotic learning in research and educational settings.

## 1 Introduction

As the field of artificial intelligence continues to evolve, robotics emerges as a fascinating area for deploying and evaluating machine learning algorithms in dynamic, real-life settings [14, 39, 46]. These applications allow embodied agents to interact within complex environments, similar to humans and animals, they must navigate a variety of challenging constraints during their learning process.

Reinforcement learning (RL), in particular, has emerged as a promising approach to learning complex behavior with robots [20, 29]. Despite the rich potential for innovation, the learning process of algorithms under real-world conditions is a challenge [1, 38, 46]. The complexity of setting up a robotics lab, combined with the high cost of equipment and the steep learning curve in RL, often limits the ability of researchers, educators, and hobbyists to contribute to and benefit from cutting-edge developments. To address these challenges, we introduce BricksRL, a comprehensive open-source framework designed to democratize access to robotics and RL. BricksRL builds upon Pybricks [44], a versatile Python package for controlling modular LEGO robotics hub, motors and sensors, actively maintained and supported by a vibrant community, and TorchRL [6], a modern framework for training

38th Conference on Neural Information Processing Systems (NeurIPS 2024).

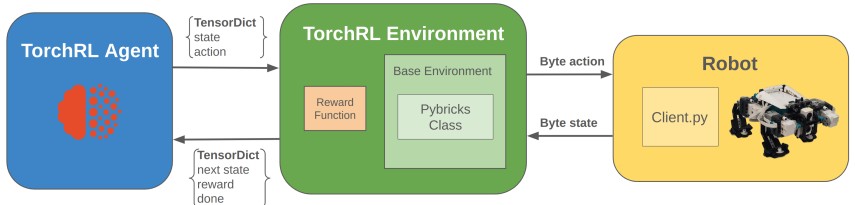

Figure 1: Communication overview of the agent, environment and robot.

RL agents. This synergy provides an integrated solution that simplifies the creation, modularity, design, and training of custom robots in the real world.

The use of LEGO parts as the basic building blocks for constructing the robots for BricksRL has the advantage of being cheap and widely available, which facilitates entry, but also makes it easier to replace parts in the event of repairs and keeps costs down. In addition, the building blocks allow full reusability of all parts, which is not the case with other robots, and no special tools are required for construction or maintenance, which also keeps costs very low. By abstracting the complexities of robot programming given a gym-like interface and RL algorithm implementation, BricksRL opens the door for a broader audience to engage with robotics research and education, making it more accessible to researchers, educators, and hobbyists alike. The low cost, wide availability, and ease of deployment also allow the introduction and use of BricksRL as a new artificial intelligence benchmark to test and evaluate RL algorithms in robotics for a variety of tasks and robots.

The contributions of our work with BricksRL are threefold. First, we provide a unified platform integrating Pybricks and TorchRL within BricksRL, which facilitates the physical control of cost-effective robotic systems and the design of gym-like training environments for RL algorithms. This setup enables scalable training across a diverse array of robots, tasks, and algorithms. Second, we demonstrate how to extend the capabilities of BricksRL beyond the standard sensor set provided by LEGO. By integrating a camera sensor, we expand the platform's capabilities, allowing for the creation of more diverse robots and tasks and thereby broadening its potential applications. Third, we present preliminary results that underscore the framework's robustness and adaptability for real-world robotics applications. Furthermore, we provide explicit examples of sim2real transfer and the application of datasets with offline RL, demonstrating the practical utility of BricksRL in robotics and RL research. We make the source code available at: `https://github.com/BricksRL/bricksrl`. The building plans, and evaluation videos of the experiments are publicly available at `https://bricksrl.github.io/ProjectPage/`.

## 2 Related Work

High acquisition costs, ranging from 10,000$ to over 200,000$, pose significant barriers to robotics research. This pricing affects various robotics types, including robotic arms (e.g., Franka [31], Kuka [23] and advanced robotic hands [29, 37, 41], similar to costly quadruped or humanoid robots designed for locomotion [5, 18, 42].

The popularization and increased consumer accessibility of 3D printing and DIY projects have heightened interest in low-cost robotics, thereby broadening access and facilitating the entry into robotics [1–3, 7, 9–11, 22, 32], lowering the initial costs for simple quadrupeds starting at 300$ robotic arms and hands for 20,000$. However, there is a requisite need for access to a 3D printer, a workshop, and equipment for the construction, maintenance, and repair of these robots. Additionally, projects and companies have been established to cater to the niche of low-cost robotics with pre-built robots for educational purposes that fall within a similar price range [4, 33–35]. Nevertheless, similar to off-the-shelf industrial robots, these and DIY robots are static, and it is not assured that printed parts or other components can be repurposed for different robots or adapted for new tasks. This limitation often confines experiments to a single robot and setting, which can be considered restrictive.

LEGO parts provide standardized and robust components that facilitate simple reconstruction, modularity of designs, and reproducibility. This modularity enables the construction and prototyping of various robots and the adaptation to different tasks, thereby simplifying the testing and benchmarking of RL algorithms across diverse robotic configurations. The initial cost for a starter kit to construct

robots starts at approximately 400$ [40], and can be augmented with additional sets, specific elements, or sensors as required. [36] demonstrates the application of LEGO for constructing robots for under 300$. However, using aluminum extrusions and 3D printed components, coupled with control via a Raspberry Pi rather than LEGO's internal PrimeHub, diminishes the system's flexibility.

In contrast, the robots in BricksRL use only LEGO elements for construction and control. The simple integration of additional sensors is demonstrated but not necessary. Further, users interact with the robots via a gym-like environment as is common in RL, simplifying the interaction. In comparison, the industry standard for managing robots and sensors is the Robotics Operating System (ROS) [33]. It offers numerous tools, however, its steep learning curve can be a barrier for researchers, students, hobbyists, and beginners starting with RL and robotics.

The use of LEGO for education in robotics has a rich history through sets such as MINDSTORMS [26] or education sets [19]. These are used not only in official educational institutions [25, 26, 43] but also in annual competitions around the world [17, 45] that attract a substantial number of children and students. To the best of our knowledge, these competitions do not currently incorporate machine learning techniques such as RL. Being tailored to these groups, BricksRL could bridge that gap and provide easy access to state-of-the-art algorithms.

## 3 BricksRL

The underlying architecture of BricksRL has three main components: the agent, the environment, and the robot 1. TorchRL is utilized to develop the agent and the environment, while Pybricks is employed for programming on the robot side. In the following sections, we will examine each component individually and discuss the communication mechanisms between them.

### 3.1 Agents

BricksRL utilizes TorchRL's modularity to create RL agents. Modules such as replay buffers, objective functions, networks, and exploration modules from TorchRL are used as building blocks to create agents that enable a uniform and clear structure. For our experiments and to showcase the integration of RL algorithms within BricksRL, we have selected three state-of-the-art algorithms for continuous control: TD3, SAC, and DroQ [12, 15, 16]. We primarily chose these off-policy algorithms for their simplicity and their proven ability of sample-efficient training, which is essential as we mostly train our robots online. However, due to the flexible and modular structure of TorchRL, BricksRL can be easily adapted to include any algorithm developed within or compatible with the TorchRL framework, allowing us to emphasize the general applicability of our system rather than the specific strategies. For example, methods commonly used in robotics, such as imitation learning [47] and the use of foundation models [28], are available with TorchRL and can be seamlessly integrated into BricksRL.

### 3.2 Environment

BricksRL incorporates TorchRL's environment design principles among other components, to standardize the structure and organization of its environments.

**PybricksHub Class.** Developed by BricksRL, the PybricksHub class plays an important role in facilitating communication with the LEGO Hub, which controls the robots. It achieves this through Bluetooth Low Energy (BLE) technology, which enables efficient, low-power communication. This class is designed to manage a two-way data exchange protocol, critical for meeting the real-time requirements of interactive applications. Importantly, the PybricksHub class seamlessly bridges asynchronous communication with the traditionally synchronous structure of RL environments.

**EnvBase.** In BricksRL, environments are designed as classes that inherit from EnvBase provided by TorchRL, which is a foundational class for building RL environments. This structure gives access to the TorchRL ecosystem and simplifies the creation of environments. Users can create custom environments or extend existing environments for new tasks. All that needs to be done is to adapt the observation and action specifications and, if necessary, define a new reward function and adapt the step and reset function of the environment.

A key advantage of using TorchRL's EnvBase in BricksRL is the ability to apply environment transforms, a fundamental feature of TorchRL. These transforms enable simple manipulation of the data exchanged between the environment and the agent. TorchRL provides a wide range of useful transforms, such as frame stacking, observation normalization, and image transformations, which are particularly valuable for real-world robotics. Additionally, the integration of foundation models like VIP [28] through transforms expands the experimentation capabilities within BricksRL. Detailed descriptions of the environments we implemented for our experiments, along with a template for creating custom environments, can be found in A.2 and A.2.8, respectively.

**Agent-Environment Communication.** For communication and data exchange between agent and environment, BricksRL makes use of TensorDict [6] as a data carrier. TensorDict enables a standardized and adaptable transmission of information between agent and environment. TensorDict can handle a wide range of data as observation by accurately describing the observation specs in the environment, without modifying the agent's or environment's essential structure. It enables users to shift between vector and picture observations or a combination of the two. This is a crucial building component for being flexible to train different algorithms on robots performing a variety of tasks with and without sensors of the LEGO ecosystem.

### 3.3   LEGO Robots

In our experiments demonstrating the capabilities of BricksRL, we selected three distinct robot types to serve as an introductory platform for RL in robotics. These robots vary in complexity and their capacity for environmental interaction, reflecting a progressive approach to robotic research. Additionally, we incorporated various sensors and embraced a range of robot classifications, showcasing a broad spectrum of applications and use cases.

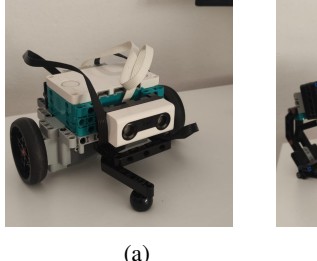
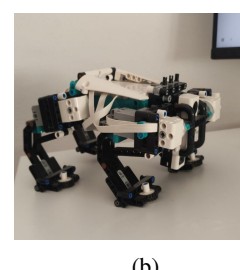
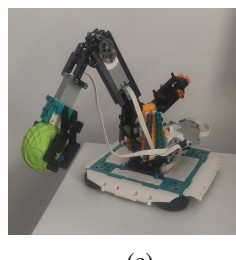

(a)                    (b)                    (c)

Figure 2: Three robots that we used in the experiments: (a) 2Wheeler, (b) Walker, (c) RoboArm.

**2Wheeler.** The 2Wheeler robot 2a is built by us to represent an elementary robotic platform designed for introductory purposes, incorporating the LEGO's Hub, a pair of direct current (DC) motors equipped with rotational sensors, and an ultrasonic sensor for determining the proximity to objects. The independent control capability of the DC motors endows the robot with high maneuverability.

**Walker.** The Walker 2b, a quadrupedal robot as built in a standard LEGO robotics kit, is equipped with four motors, each integrated with rotational sensors and an additional ultrasonic sensor. In comparison to the 2Wheeler robot 3.3, the Walker variant exhibits more degrees of freedom and a higher level of complexity due to its increased motor count and the fact that it uses legs instead of wheels. In terms of structural design, this robot bears similarity to prevalent quadruped robots typically employed in the domain of locomotion control [39].

**RoboArm.** The RoboArm 2c is built by us and is similar to the Walker 4 motors with rotation sensors equipped, however, it has a higher range of motion. Further, it is the only static robot and includes another branch of robot types used for tasks like grasping and reaching or manipulating objects [20, 21, 27].

In general, Pybricks allows wide access to different motors and sensors as well as IMU (Inertial Measurement Unit) data in the LEGO's hub, which permits a variety of possible modular robot architectures and applications. For a detailed overview, please refer to the official Pybricks documentation

[30]. Furthermore, we highlight the large community that has already created various robots, which can be rebuilt or used as inspiration. Any robot can be used with BricksRL as long as it is available in Pybricks's interface. We welcome collaborations and encourage community members who want to experiment with BricksRL to contact us for support.

**Robot-Environment Communication.**  BricksRL employs a bidirectional data flow between the robot and the environment, facilitated by MicroPython in Pybricks. The agent's actions are transmitted as byte streams via standard input (stdin), where they are parsed and applied to the robot's motors. At the same time, the robot sensor data is sent back to the environment through standard output (stdout) for state evaluation and action selection. Each robot uses a dedicated client script to manage its motors, sensors, and control flow for specific tasks. For exact details and an example of a typical client script, see the provided template in A.3.

**Communication Speed.**  In robotics and motion control, the rate of communication is crucial, necessitating high frequencies to ensure rapid responsiveness and precision in response to environmental changes or disturbances. Position or torque-based control systems in quadrupedal robots, for instance, operate within a query frequency range of 20 to 200 Hz [8, 24]. This frequency range enables these robots to swiftly adjust to variations in terrain during locomotion.

Likewise, the hub's control loop is capable of exceeding frequencies of 1000 Hz, making it suitable for managing complex robotic systems. Yet, when integrating with the BricksRL communication framework, a reduction in the system-wide frequency, including agent-environment and environment-robot communications, to 11 Hz was observed. This decrease is primarily due to the overhead introduced by utilizing stdin and stdout for communication. The process of reading from and writing to these streams, which requires system calls, is inherently slower compared to direct memory operations. Additionally, the necessity to serialize and deserialize data through 'ustruct.unpack' and 'ustruct.pack' adds to this overhead, as it requires converting data between binary formats used in communication and the Python object representation, which is time-consuming.

Despite the overhead, BricksRL's communication speed, while on the lower spectrum, remains within a reasonable range for robotic system applications. For instance,[13, 39] have demonstrated that effective motion control in quadrupedal robots can be achieved at much lower frequencies, such as 20 or even 8 Hz, indicating that robust and dynamic locomotion can be maintained even at reduced communication frequencies. Moreover, we show in our experiments that communication frequency is task and robot depending, for specific tasks optimal behaviors can be learnt faster with lower frequencies 7.

### 3.4  Modularity and Reusability

The use of interlocking LEGO parts, and various sensors allows for endless possibilities in designing and building robots and robot systems. Additionally, precise construction plans and the reusability of components create additional opportunities. Unlike classic robot systems, users are not limited to a single design and functionality. Instead, robots can be customized to specific requirements or tasks, and can even be reassembled for new challenges or ideas once the initial task is completed successfully. Building on this variable foundation with an infinite number of robotic applications, BricksRL enables easy interaction by abstracting the complexities of the underlying communication processes. To train RL algorithms, users can interact with the robots in gym-like environments, which provide a natural and intuitive interface. This enables researchers and hobbyists to train any RL algorithm, such as on-policy or off-policy, model-based or model-free.

To further illustrate modularity and scalability of BricksRL we extended the set of sensors and show how easy it is to integrate sensors outside of the LEGO ecosystem. Namely, we integrate a USB webcam camera into an environment (A.2.7) showcased in the experiments, demonstrating that additional sensors can further augment the scope of applications to train robots with RL and BricksRL.

## 4  Experiments

In our experiments, we aim to address several critical questions regarding the feasibility and efficiency of training LEGO robots using RL algorithms in the real world with BricksRL. Thereby taking into

account the practical challenges of training LEGO robots such as the lack of millimeter-precise robot constructions, the presence of backlash, and noisy sensors.

| Robot | Environments |
|---|---|
| 2Wheeler | RunAway-v0 |
| | Spinning-v0 |
| Walker | Walker-v0 |
| | WalkerSim-v0$^\dagger$ |
| Roboarm | RoboArm-v0 |
| | RoboArmSim-v0$^\dagger$ |
| | RoboArm-mixed-v0* |

Table 1: Overview of BricksRL Robot and Environment Settings. Environments marked with an asterisk (*) utilize LEGO sensors and image inputs as observations for the agent. Environments indicated by a dagger ($\dagger$) denote simulations of the real robot and do not use the real robot for training.

Therefore we developed various task-specific environments to demonstrate the adaptability and ease of use of BricksRL, highlighting the scalability of training across different algorithms and robots with diverse sensor arrays. Tasks ranging from driving and controlling the 2Wheeler to learning to walk with the Walker and reaching tasks for the RoboArm demonstrate the applicability of BricksRL. Table 1 shows a complete overview of all environments used in our experiments.

In our experiments, we primarily focus on online learning, where the robot directly interacts with the real world, encompassing the challenges inherent to this approach. However, we have also developed simulation environments for certain tasks. Training in these simulations is significantly faster compared to real-world training, as confirmed by our comparative experiments. Additionally, we use these simulation environments to demonstrate the sim2real capabilities of LEGO robots with BricksRL.

A complete overview and description of the environments implemented including action and observation specifications as well as the definition of the reward function can be found in the appendix A.2. We also provide an environment template 1 that demonstrates the straightforward process of creating custom environments using BricksRL.

In all of our experiments, we initiated the training process with 10 episodes of random actions to populate the replay buffer. The results are obtained by over 5 seeds for each algorithm and compared against a random policy. Evaluation scores of the trained policies are displayed in 2. We further provide videos of trained policies for each robot and task A.1. Hyperparameter optimization was not conducted for any of the algorithms, and we adhered to default settings. Comprehensive information on the hyperparameter is provided in the appendix 12. Although the option to utilize environment transformations, such as frame stacking and action repetition, was available, we opted not to use these features to maintain the simplicity of our setup. Further details are available in the appendix A.2.

| | Environment | | | | | | |
|---|---|---|---|---|---|---|---|
| Algorithm | RunAway-v0 | Spinning-v0 | Walker-v0 | WalkerSim-v0 | RoboArm-v0 | RoboArmSim-v0 | RoboArm-mixed-v0 |
| TD3 | $7.64 \pm 2.31$ | $\mathbf{7558.21 \pm 28.61}$ | $-62.94 \pm 16.76$ | $-78.49 \pm 9.75$ | $\mathbf{-20.29 \pm 31.35}$ | $-12.78 \pm 21.09$ | $-60.24 \pm 16.06$ |
| SAC | $8.72 \pm 2.82$ | $7407.20 \pm 109.53$ | $\mathbf{-52.04 \pm 8.79}$ | $\mathbf{-55.03 \pm 4.95}$ | $-27.77 \pm 37.13$ | $\mathbf{-3.45 \pm 2.66}$ | $\mathbf{-18.21 \pm 6.98}$ |
| DroQ | $\mathbf{8.96 \pm 0.87}$ | $7456.85 \pm 18.02$ | $-57.63 \pm 10.44$ | $-56.62 \pm 2.81$ | $-55.02 \pm 62.45$ | $-14.04 \pm 26.05$ | $-19.39 \pm 11.07$ |
| Random | $-0.51 \pm 1.84$ | $71.97 \pm 501.79$ | $-191.99 \pm 18.19$ | $-191.99 \pm 18.19$ | $-149.26 \pm 88.19$ | $-149.26 \pm 88.19$ | $-57.23 \pm 10.01$ |

Table 2: The table displays the mean and standard deviation of evaluation rewards for the trained TD3, SAC, DroQ algorithms, and a random policy, based on experiments conducted across 5 evaluation episodes and 5 different seeds.

## 4.1 2Wheeler

In the `RunAway-v0` task for the 2Wheeler robot, we trained RL algorithms over 40 episodes. Training sessions were completed in approximately 15 minutes per run for each agent. All algorithms successfully mastered the task, as shown in Figure 3. Notably, despite the simplicity of the task, algorithms adopted unique strategies. TD3 maximized its actions, achieving the highest distance from the wall but causing rapid acceleration, tilting, and noisy measurements, leading to occasional

abrupt episode termination. In contrast, the DroQ agent used smaller actions, resulting in more stable but shorter distances and avoiding premature episode endings A.5.

For the `Spinning-v0` task, we trained the agents over 15 episodes. The training was completed in about 10 minutes, with all agents effectively learning to solve the task, as illustrated in Figure 3. Table 2 includes the evaluation scores for both tasks of the 2Wheeler.

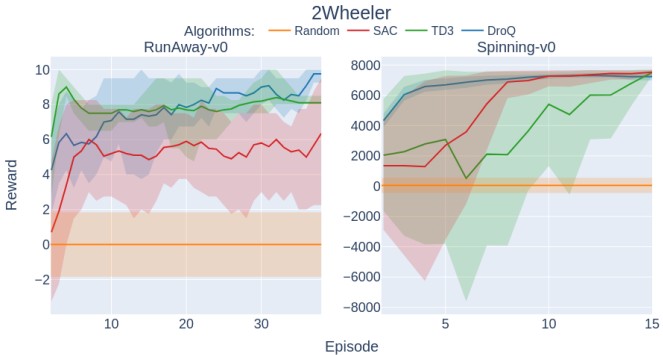

Figure 3: Training results for 2Wheeler robot for the `RunAway-v0` and the `Spinning-v0` environment.

## 4.2 Walker

In the `Walker-v0` environment, we trained the Walker robot over 75 episodes. The entire training process took approximately 70 minutes. All agents successfully developed a forward-moving gait. Remarkably, the DroQ algorithm achieved this in significantly fewer episodes (5-10), requiring only about 15 minutes of training, as illustrated in Figure 4. We trained the agents with a communication frequency of 2 Hz, instead of the maximum frequency of 11 Hz, as we observed that training at the lower frequency was faster and more stable. Results of a direct comparison can be found in the appendix 7.

Figure 4 also presents results from training in the `WalkerSim-v0` environment. To be comparable with the `WalkerSim-v0`, we similarly trained the agents for 75 episodes but noted marked differences in training duration: TD3 and SAC completed within 1-3 minutes, whereas DroQ required about 20 minutes due to a higher updating ratio. Simulation results revealed higher training performance with smoother and more stable learning curves compared to real-world training 4.

The final evaluation scores for policies trained in both real-world and simulation environments, tested on the actual robot, are summarized in Table 2. Although the simulation-trained policies slightly underperformed, they demonstrated effective sim2real transfer, highlighting their efficiency with considerably less training time and reduced supervision.

|           | Success Rate (%) | |
| --------- | ---------- | ---------------- |
| Algorithm | RoboArm-v0 | RoboArm-mixed-v0 |
| TD3       | 88         | 8                |
| SAC       | 72         | 68               |
| DroQ      | 64         | 68               |
| Random    | 32         | 40               |
| TD3*      | 88         | -                |
| SAC*      | 100        | -                |
| DroQ*     | 88         | -                |

Table 3: Comparison of success rates for different agents in the `RoboArm-v0` and `RoboArm-mixed-v0` environments. Success is defined as the agent reaching the goal or goal position within a specified threshold. Agents marked with an asterisk (*) were initially trained in the `RoboArmSim-v0` environment. Each algorithm was evaluated for 5 epochs with 5 different seeds, totaling 25 experiments per agent and task.

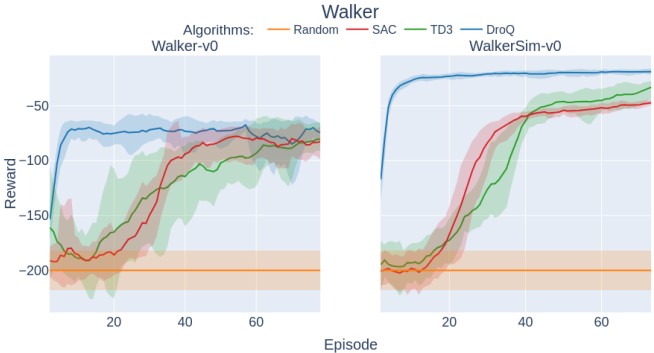

Figure 4: Training performance for Walker robot for the `Walker-v0` and the `WalkerSim-v0` environment.

## 4.3 RoboArm

In the `RoboArm-v0` task, agents were trained for 250 episodes. Training durations varied, with TD3 and SAC completing in about 1 hour on the real robot, whereas DroQ required close to 2 hours. By contrast, training in the `RoboArmSim-v0` environment proved much quicker: TD3 and SAC finished within 1-2 minutes, and DroQ in approximately 25 minutes. The outcomes, depicted in Figure 5, confirm that all agents successfully learned effective policies.

To enhance the interpretation of the training outcomes, we also plotted the final error—defined as the deviation from the target angles at the last step of each episode—and the total number of steps taken per episode. The data reveals a consistent decrease in both the final error and the number of steps throughout the training period. This indicates not only improved accuracy but also increased efficiency, as episodes terminated sooner when goals were successfully met.

The evaluation results are detailed in Table 2. Additionally, we compiled success rates that illustrate how often each agent reached the goal position within the predefined threshold. These success rates, derived from evaluation runs across 5 seeds with each seed running 5 episodes, are presented in Table 3. Notably, the policies trained in the `RoboArmSim-v0` environment achieved superior evaluation scores and also higher success rates upon testing. This demonstrates a successful sim2real transfer, achieving a significantly reduced training time.

Lastly, we present the training results for the `RoboArm_mixed-v0` environment, where the algorithms underwent training over 250 episodes. Training durations varied significantly due to the complexity added by integrating additional image observations: SAC was completed in 40 minutes, TD3 in 60 minutes, and DroQ took three hours. The inclusion of image data likely introduced considerable noise in the training results, as illustrated in Figure 6, which displays the rewards achieved by the agents and the corresponding episode steps. Interestingly, while SAC and DroQ successfully learned effective policies, TD3 struggled to adapt, failing to develop a viable strategy. The success of SAC and DroQ is evident in the chart of episode steps, showing a decrease in steps over the training period, which indicates a more efficient achievement of the goal position.

The evaluation results, detailed in Table 2, confirm the performances. Notably, out of 25 evaluation trials, both SAC and DroQ successfully reached the goal position 17 times, as recorded in Table 3. This demonstrates the robustness of the SAC and DroQ algorithms in handling the complexities introduced in the `RoboArm_mixed-v0` environment.

## 4.4 Offline Training

Offline RL uses pre-collected datasets to train algorithms efficiently, avoiding real-world interactions and complex simulations. To further highlight the capabilities of BricksRL for education and research in robotics and RL we collected offline datasets for the LEGO robots. With those datasets, BricksRL allows training of the LEGO robots via offline RL or imitation learning, both of which are state-of-the-art methods for RL in robotics.

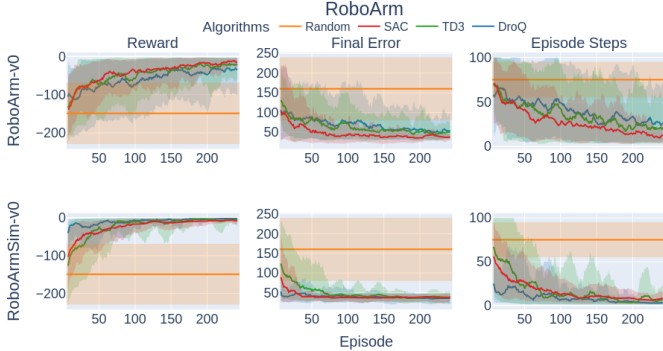

Figure 5: Training outcomes for the RoboArm robot in both the `RoboArm-v0` and `RoboArmSim-v0` environments. The plot also includes the final error at the epoch's last step and the total number of episode steps.

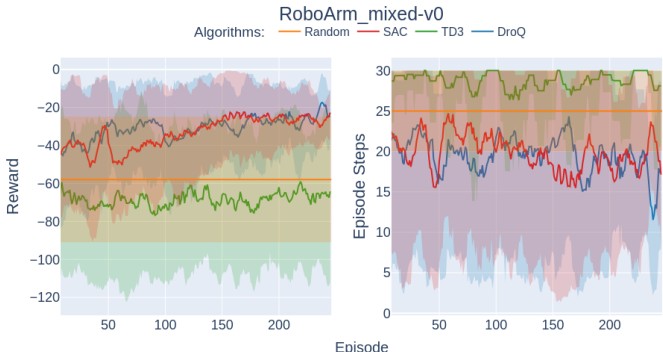

Figure 6: Training performance of the RoboArm robot in the `RoboArm_mixed-v0` environment, showing both the reward and the number of episode steps required to reach the target location.

For BricksRL, we curated datasets for three robot configurations: 2Wheeler, Walker, and RoboArm. These datasets include both expert and random data for four tasks in our experiments (Walker-v0, RoboArm-v0, RunAway-v0, Spinning-v0). Details about the datasets and dataset generation can be found in the appendix A.7. Using these datasets, we demonstrated that offline RL with BricksRL is feasible, successfully training both online and offline RL algorithms and applying them to a real robot. The evaluation performance, shown in Table 4, highlights the superior performance of offline RL algorithms, particularly with expert data, while online algorithms struggle, suggesting overfitting or poor generalization. For further details on training parameters, please refer to the appendix A.8.

| Agent | Walker-v0 | | RoboArm-v0 | | RunAway-v0 | | Spinning-v0 | |
| | Random | Expert | Random | Expert | Random | Expert | Random | Expert |
|---|---|---|---|---|---|---|---|---|
| TD3 | $-79.71$ | $-153.45$ | $-124.22$ | $-201.94$ | **19.86** | $9.06$ | **6160.25** | $6168.28$ |
| SAC | $-66.91$ | $-255.41$ | $-54.67$ | $-218.76$ | $14.74$ | $10.80$ | $5416.11$ | **9349.52** |
| BC | $-202.46$ | $-85.65$ | $-117.72$ | $-7.34$ | $-0.27$ | $18.13$ | $35.55$ | $9150.02$ |
| IQL | $-136.13$ | **$-74.80$** | $-76.89$ | **$-3.10$** | $14.07$ | $18.80$ | $4544.28$ | $9096.60$ |
| CQL | $-72.75$ | $-77.93$ | **$-46.91$** | $-17.41$ | $19.60$ | **19.74** | $4509.31$ | $9099.03$ |

Table 4: Evaluation Results: Online (TD3, SAC) and Offline (BC, IQL, CQL) RL Algorithms. Scores represent the mean reward averaged over 5 episodes and 5 random seeds.

# 5 Conclusion

In this paper, we introduce BricksRL and detail its benefits for robotics, RL, and educational applications, emphasizing its cost-effectiveness, reusability, and accessibility. In addition, we showcased its practical utility by deploying three distinct robots, performing various tasks with a range of sensors, across more than 100 experiments. Our results underscore the viability of integrating state-of-the-art RL methodologies through BricksRL within research and educational contexts. By providing comprehensive building plans and facilitating access to BricksRL, we aim to establish this investigation as a foundational proof of concept for utilizing LEGO-based robots to train RL algorithms. Moving forward, avenues for further research include creating more complex robots and tasks, exploring applications in multi-agent settings, and leveraging large datasets to enhance RL training through transformer-based imitation learning. Ultimately, BricksRL sets the stage for a future where accessible, reusable robotic systems support and expand RL research, collaborative learning, and interactive education.

## Acknowledgements

We thank A. De Fabritiis Campos, M. De Fabritiis Campos and P. Vallecillos Cusco for providing their LEGOs to this project.

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

## A  Appendix

### A.1  Repository and Website

BricksRL repository and our project website with evaluation videos and building instructions can be found at the following locations:

- GitHub: BricksRL
- Project Website

### A.2  Environments

#### A.2.1  RunAway-v0

The `RunAway-v0` environment presents a straightforward task designed for the 2Wheeler robot. The objective is to maximize the distance measured by the robot's ultrasonic sensor. To accomplish this, the agent controls the motor angles, deciding how far to move forward or backward. This environment operates within a continuous action space, and each episode spans a maximum of 20 steps. Episodes will terminate early if the robot reaches the maximum distance of 2000 mm.

**Action and Observation Specifications.**  TorchRL's `BoundedTensorSpecs` are employed to define the action and observation specifications, which have 1 and 5 dimensions, respectively. Table 5 outlines the specific ranges. Actions are initially defined in the range of $[-1, 1]$ but are linearly mapped to the range of $[-100, 100]$ before being applied to both the left and right wheel motors.

| Type | Num | Specification | Min | Max |
|---|---|---|---|---|
| Action Spec | 0 | motor | -1 | 1 |
| Observation Spec | 0 | left motor angle | 0.0 | 360.0 |
| | 1 | right motor angle | 0.0 | 360.0 |
| | 2 | pitch angle | -90 | 90 |
| | 3 | roll angle | -90 | 90 |
| | 4 | distance | 0.0 | 2000.0 |

Table 5: Combined action and observation specifications for the `RunAway-v0` environment.

**Reward Function.**  The reward function for the RunAway-v0 environment is defined as:

$$R_t = \begin{cases} +1 & \text{if distance}_t > \text{distance}_{t-1} \\ -1 & \text{if distance}_t < \text{distance}_{t-1} \\ 0 & \text{else} \end{cases} \tag{1}$$

### A.2.2 Spinning-v0

The `Spinning-v0` environment is another setup designed for the 2Wheeler robot. Unlike `RunAway-v0`, this environment does not use the ultrasonic sensor to measure the robot's distance to objects in front of it. Instead, at each reset, a value is randomly selected from the discrete set $0, 1$, which explicitly dictates the rotational direction in which the robot should spin. The robot's IMU (Inertial Measurement Unit) enables the tracking of various parameters, including angular velocity. This angular velocity is part of the agent's observation in the `Spinning-v0` environment, providing it with information on its rotational direction to complete the task. The action space is also continuous, and each episode has a length of 50 steps.

**Action and Observation Specifications.** The `BoundedTensorSpec` for the observation in the `Spinning-v0` environment comprises six floating-point values: the left and right motor angles, the pitch and roll angles, the angular velocity $\omega_z$, and the rotational direction. Table 6 details these specifications. The action specification is defined as two floating-point values representing the rotation angles applied to the left and right motors. Actions are initially defined in the range of $[-1, 1]$ but will be transformed to the range of $[-100, 100]$ before being applied to the motors, as detailed in Table 6.

| Type | Num | Specification | Min | Max |
|---|---|---|---|---|
| Action Spec | 0 | left motor | -1 | 1 |
| | 1 | right motor | -1 | 1 |
| Observation Spec | 0 | left motor angle | 0.0 | 360.0 |
| | 1 | right motor angle | 0.0 | 360.0 |
| | 2 | pitch angle | -90 | 90 |
| | 3 | roll angle | -90 | 90 |
| | 4 | angular velocity $\omega_z$ | -100 | 100 |
| | 5 | direction | 0 | 1 |

Table 6: Combined action and observation specifications for the `Spinning-v0` environment.

**Reward Function.** The reward function for the Spinning-v0 environment is defined in 1. The angular velocity $\omega_z$ is directly used as a reward signal and encourages adherence to a predefined rotational orientation.

$$R_t = \begin{cases} \omega_{z,t} & \text{if rotational direction} = 0 \text{ (spinning left)}, \\ -\omega_{z,t} & \text{otherwise (spinning right)}. \end{cases} \tag{2}$$

### A.2.3 Walker-v0

In the `Walker-v0` environment for the Walker robot, the objective is to master forward movement using its four legs. To achieve this, the robot is provided with data on the current angles of each leg's motors, along with IMU readings that include both pitch and roll angles, ensuring operational safety. Additionally, an ultrasonic sensor is used to momentarily halt the agent's actions when the detected distance falls below a predefined threshold, preventing collisions with obstacles. Each episode consists of 100 steps. To achieve reduced communication speed, a waiting time is added after the actions are applied to the motors.

**Action and Observation Specifications.** In the `Walker-v0` environment, the observation specification consists of seven floating-point values: four motor angles (one for each leg), the pitch and roll angles, and the distance measurements from the ultrasonic sensor. The action specification includes four floating-point values corresponding to the four leg motors. These action values are initially defined in the range of $[-1, 1]$ but are linearly mapped to the range of $[-100, 0]$ before being applied, as detailed in Table 7.

**Reward Function.** The total reward is a sum of penalties for the actions and differences in angles, encouraging synchronized movement and appropriate angular differences between the legs of the

| Type | Num | Specification | Min | Max |
|------|-----|---------------|-----|-----|
| Action Spec | 0 | left front motor | -1 | 1 |
| | 1 | right front motor | -1 | 1 |
| | 2 | left back motor | -1 | 1 |
| | 3 | right back motor | -1 | 1 |
| Observation Spec | 0 | left front motor angle | 0.0 | 360.0 |
| | 1 | right front motor angle | 0.0 | 360.0 |
| | 2 | left back motor angle | 0.0 | 360.0 |
| | 3 | right back motor angle | 0.0 | 360.0 |
| | 4 | pitch angle | -90 | 90 |
| | 5 | roll angle | -90 | 90 |
| | 6 | distance | 0.0 | 2000.0 |

Table 7: Combined action and observation specifications for the `Walker-v0` environment.

walker. The reward components are defined as follows: - $R_{\text{action},t}$: Penalty for the magnitude of actions taken. - $R_{\text{lf-rb},t}$: Penalty for the angular difference between the left front (lf) and right back (rb) motor angles. - $R_{\text{rf-lb},t}$: Penalty for the angular difference between the right front (rf) and left back (lb) motor angles. - $R_{\text{lf-rf},t}$: Penalty for the deviation from 180 degrees between the left front (lf) and right front (rf) motor angles. - $R_{\text{lb-rb},t}$: Penalty for the deviation from 180 degrees between the left back (lb) and right back (rb) motor angles.

$$\mathbf{R}_t = R_{\text{action},t} + R_{\text{lf-rb},t} + R_{\text{rf-lb},t} + R_{\text{lf-rf},t} + R_{\text{lb-rb},t} \tag{3}$$

$$R_{\text{action},t} = -\frac{\sum \text{actions}_t}{40} \tag{4}$$

$$R_{\text{lf-rb},t} = -\frac{\text{angular\_difference}(\theta_{\text{lf},t}, \theta_{\text{rb},t})}{180} \tag{5}$$

$$R_{\text{rf-lb},t} = -\frac{\text{angular\_difference}(\theta_{\text{rf},t}, \theta_{\text{lb},t})}{180} \tag{6}$$

$$R_{\text{lf-rf},t} = -\frac{180 - \text{angular\_difference}(\theta_{\text{lf},t}, \theta_{\text{rf},t})}{180} \tag{7}$$

$$R_{\text{lb-rb},t} = -\frac{180 - \text{angular\_difference}(\theta_{\text{lb},t}, \theta_{\text{rb},t})}{180} \tag{8}$$

with the angular difference defined as:

$$\text{angular\_difference}(\theta_{1,t}, \theta_{2,t}) = |((\theta_{2,t} - \theta_{1,t} + 180) \mod 360) - 180| \tag{9}$$

### A.2.4 WalkerSim-v0

Additionally, we have developed a simulated version of the `Walker-v0` environment, called `WalkerSim-v0`. This simulation mirrors the real-world setup without requiring communication with the actual robot or using PyBricks. In the simulation, both IMU measurements and ultrasonic sensor inputs, which are used in the `Walker-v0` environment, are set to zero. This simplification is made because modeling or simulating these sensors can be challenging due to their complexity and the nuances involved in accurately replicating their readings. However, since these sensors are primarily used for safety in the real world, their absence is not a concern in the simulated environment.

In `WalkerSim-v0`, the next motor states are calculated by simulating the transition dynamics. This involves transforming the action output into the angle range and then applying the corresponding actions by adding to the current motor state. To model real-world inaccuracies, we add noise to the motor states, sampled from a Gaussian distribution with a mean of 0 and a standard deviation of 0.1. Similar to the real-world environment the `WalkerSim-v0` episodes consist of 100 interactions.

**Action and Observation Specifications.** Action and observation specifications are the same as in the `Walker-v0` 7.

**Reward Function.** The reward function for the `WalkerSim-v0` is the same as in the `Walker-v0` environment.

### A.2.5 RoboArm-v0

The `RoboArm-v0` is a pose-reaching task. At every reset, random goal angles for the four motors are sampled, defining a target pose. The objective is to adjust the robot's articulation to reach the specified goal pose within 100 steps. To achieve this the robot is provided with the current motor angles of all joints. The design of this environment permits the user to choose whether the robot tackles the task with dense or sparse rewards, effectively adjusting the difficulty level.

**Action and Observation Specifications.** The observation specification for the `RoboArm-v0` environment consists of eight floating-point values: four current motor angles and four goal motor angles, as detailed in Table 8. The action specification is defined by four floating-point values for the four motors, initially in the range of $[-1, 1]$. Before applying the specific actions to each motor, these values are transformed as follows: the rotation motor actions are linearly mapped to the range $[-100, 100]$, the low motor actions to $[-30, 30]$, the high motor actions to $[-60, 60]$, and the grab motor actions to $[-25, 25]$.

| Type | Num | Specification | Min | Max |
|---|---|---|---|---|
| | 0 | rotation motor | -1 | 1 |
| | 1 | low motor | -1 | 1 |
| Action Spec | 2 | high motor | -1 | 1 |
| | 3 | grab motor | -1 | 1 |
| | 0 | rotation motor angle | 0.0 | 360.0 |
| | 1 | low motor angle | 10 | 70 |
| | 2 | high motor angle | -150 | 10 |
| | 3 | grab motor angle | -148 | -45 |
| Observation Spec | 4 | goal rotation motor angle | 0 | 360 |
| | 5 | goal low motor angle | 10 | 70 |
| | 6 | goal high motor angle | -150 | 10 |
| | 7 | goal grab motor angle | -148 | -45 |

Table 8: Combined action and observation specifications for the `RoboArm-v0` environment.

**Reward Function.** The reward function for the `RoboArm-v0` environment can be chosen to be either dense or sparse. In the sparse case, the reward is 1 if the distance between the current motor angles and the goal motor angles is below a defined threshold; otherwise, it is 0. In our experiments, however, we used the dense reward function, which is calculated as follows:

$$R_t = -\frac{\|\Delta\vec{\theta}_{\text{deg},t}\|_1}{100} \tag{10}$$

where $\Delta\vec{\theta}_{\text{deg},t}$ is the vector of shortest angular distances at time step $t$ between the goal motor angles $\vec{\theta}_{\text{goal},t}$ and the current motor angles $\vec{\theta}_{\text{current},t}$, defined as:

$$\Delta\vec{\theta}_{\text{deg},t} = \text{degrees}\Big( \arctan 2 \Big( \sin(\text{radians}(\vec{\theta}_{\text{goal},t}) - \text{radians}(\vec{\theta}_{\text{current},t})),$$
$$\cos(\text{radians}(\vec{\theta}_{\text{goal},t}) - \text{radians}(\vec{\theta}_{\text{current},t})) \Big) \Big) \tag{11}$$

### A.2.6 RoboArmSim-v0

`RoboArmSim-v0` mirrors the real-world `RoboArm-v0` environment but is entirely simulated, removing the need for physical interaction with an actual robot or the use of PyBricks. In this virtual setup, the RoboArm's task remains a pose-reaching challenge, where it must align its articulation to match randomly sampled goal angles for its four motors, setting a target pose at every reset. The robot in the simulation is provided with the current motor angles of all joints and the goal angles to accomplish this task within 100 steps. Crucially, the simulation is straightforward since it does not require modeling or simulating complex sensor measurements for state transitions, but only the motor angles, simplifying the simulation of state transitions significantly. Similar to the `WalkerSim-v0` we add Gaussian noise ($\mathcal{N}(0, 0.05)$ ) to the actions before the linear mapping and addition with the current motor states. Like its real-world counterpart, this environment allows users to choose between dense and sparse reward structures, facilitating the adjustment of the task's difficulty level.

**Action and Observation Specifications.** Action and observation specifications are the same as in the `RoboArm-v0` 8.

**Reward Function.** In the `RoboArmSim-v0` environment we use the same dense reward function as defined for the `RoboArm-v0` environment.

### A.2.7 RoboArm-mixed-v0

In the `RoboArm_mixed-v0` environment, an additional sensor input is available to the robot through a webcam. The RoboArm holds a red ball in its hand and must move it to a target position. Each episode has a maximum of 30 steps. The target position is randomly selected and displayed as a green circle in the image. The image serves as additional information for the algorithm and is used to determine if the conditions to solve the task are met.

**Action and Observation Specifications.** The full observation specifications for the `RoboArm_mixed-v0` environment consist of three floating-point values representing the three motor angles (rotation motor, low motor, high motor) and image observation specifications with a shape of (64, 64), as detailed in Table 9.

The action specifications are also three floating-point values in the range of $[-1, 1]$, which will be transformed before being applied to the specific motor. The rotation motor angles are transformed to the range of $[-90, 90]$, the low motor angles to the range of $[-30, 30]$, and the high motor angles to the range of $[-60, 60]$.

| Type | Num | Specification | Min | Max |
|---|---|---|---|---|
| | 0 | rotation motor | -1 | 1 |
| Action Spec | 1 | low motor | -1 | 1 |
| | 2 | high motor | -1 | 1 |
| | 0 | rotation motor angle | 0.0 | 360.0 |
| Observation Spec | 1 | low motor angle | 10 | 70 |
| | 2 | high motor angle | -150 | 10 |
| Image Observation Spec | 0 | image observation Size: (64, 64) | 0 | 255 |

Table 9: Combined action and observation specifications for the `RoboArm_mixed-v0` environment.

**Reward Function.** To calculate the reward for the mixed observation environment `RoboArm_mixed-v0`, we utilize the Python package `OpenCV` to detect the red ball and measure the distance to the target location depicted in the image. First, we convert the image from BGR to HSV and define a color range to identify the contours. For each detected contour, we calculate the distance to the center of the green target circle and take the mean distance as the reward:

$$R_t = -\frac{\sum \text{distances}_t}{n_t \cdot 100} \tag{12}$$

where $n_t$ is the number of distances detected at the current time step. If no contours are detected, we use the previous reward as the current reward.

### A.2.8  Task Environment Template

To illustrate the simplicity of using the TorchBricksRL BaseEnv, which manages communication between the environment and the robot, we provide an example in Listing 1. This code can serve as a template for creating new custom environments.

```python
import torch

from environments.base.base_env import BaseEnv
from tensordict import TensorDict, TensorDictBase
from torchrl.data.tensor_specs import BoundedTensorSpec, CompositeSpec

class TaskEnvironment(BaseEnv):
    # Define your action and state dimension, needs to be adapted
    depending on your task and robot!
    action_dim = 1  # One action to control the wheel motors together
    state_dim = 5  # 5 sensors readings (left motor angle, right motor
     angle, pitch, roll, distance)

    # Define observation space ranges.
    motor_angles = [0, 360]
    roll_angles = [-90, 90]
    pitch_angles = [-90, 90]
    distance = [0, 2000]

    observation_key = "vec_observation"

    def __init__(
        self,
    ):
        self._batch_size = torch.Size([1])

        # Define Action Spec.
        self.action_spec = BoundedTensorSpec(low=-1, high=1, shape=(1,
    self.action_dim))
        # Define Observation Spec.
        bounds = torch.tensor(
            [
                self.motor_angles,
                self.motor_angles,
                self.roll_angles,
                self.pitch_angles,
                self.distance,
            ]
        )
        observation_spec = BoundedTensorSpec(
            low=bounds[:, 0],
            high=bounds[:, 1],
        )
        self.observation_spec = CompositeSpec({self.observation_key:
    observation_spec}, shape=(1,))

        super().__init__(
            action_dim=self.action_dim, state_dim=self.state_dim,
        )

```

```
49    def _reset(self, tensordict: TensorDictBase, **kwargs) ->
      TensorDictBase:
50        # Get initial state from hub.
51        observation = self.read_from_hub()
52        # Could also add external sensors here and return them as well
          .
53        # img = self.camera.read()
54        return TensorDict(
55            {
56                self.observation_key: norm_observation.float(),
57                # image_obs: img.float(),
58            },
59            batch_size=[1],
60        )
61
62    def reward(self, state, action, next_state) -> Tuple[float, bool]:
63        # Define your reward function.
64        # ...
65        reward, done = 0, False
66        return reward, done
67
68    def _step(self, tensordict: TensorDictBase) -> TensorDictBase:
69        # Send action to hub to receive next state.
70        action = tensordict.get("action").cpu().numpy().squeeze(0)
71        self.send_to_hub(action)
72        # Read next state from hub.
73        next_observation = self.read_from_hub()
74
75        # Compute the reward.
76        state = tensordict.get(self.original_vec_observation_key)
77        next_state = next_tensordict.get(self.
      original_vec_observation_key)
78        reward, done = self.reward(
79            state=state,
80            action=action,
81            next_state=next_state,
82        )
83        # Create output TensorDict.
84        next_tensordict = TensorDict(
85            {
86                self.observation_key: self.normalize_state(
      next_observation).float(),
87                "reward": torch.tensor([reward]).float(),
88                "done": torch.tensor([done]).bool(),
89            },
90            batch_size=[1],
91            device=tensordict.device,
92        )
93        return next_tensordict
```

Listing 1: Task environment template

### A.3 Client Script

For each task and robot, a custom client script is required to facilitate interaction between the robot and the environment. The client.py script defines the configuration of motors, sensors, and the workflow for processing and exchanging data. This script must be uploaded to the Pybricks Hub and updated whenever the robot's configuration changes, such as when motors or sensors are added or removed. Listing 2 provides a simple example of a client script tailored for the RunAway-v0 task. In this example, a single float value representing the action is used to control the motors, while sensor data is collected and transmitted back to the environment.

```
1  import ustruct
2  from micropython import kbd_intr
```

```python
3  from pybricks.hubs import InventorHub
4  from pybricks.parameters import Direction, Port
5  from pybricks.pupdevices import Motor, UltrasonicSensor
6  from pybricks.robotics import DriveBase
7  from pybricks.tools import wait
8  from uselect import poll
9  from usys import stdin, stdout
10
11
12 # Initialize the Inventor Hub.
13 hub = InventorHub()
14
15 # Initialize the drive base.
16 left_motor = Motor(Port.E, Direction.COUNTERCLOCKWISE)
17 right_motor = Motor(Port.A)
18 drive_base = DriveBase(left_motor, right_motor)
19 # Initialize the distance sensor.
20 sensor = UltrasonicSensor(Port.C)
21
22 keyboard = poll()
23 keyboard.register(stdin)
24
25 while True:
26
27     # Optional: Check available input.
28     while not keyboard.poll(0):
29         wait(1)
30
31     # Read action values for the motors.
32     action = ustruct.unpack("!f", stdin.buffer.read(4))[0]
33     # Apply the action to the motors
34     drive_base.straight(action, wait=True)
35
36     # Read sensors to get current state of the robot.
37     (left_m_angle, right_m_angle) = (left_motor.angle(), right_motor.
    angle())
38     (pitch, roll) = hub.imu.tilt()
39     dist = sensor.distance()
40
41     # Send the current state back to the environment.
42     out_msg = ustruct.pack(
43         "!fffff", left_m_angle, right_m_angle, pitch, roll, dist
44     )
45     stdout.buffer.write(out_msg)
```

Listing 2: Client script example.

### A.4 Communication Frequency

Figure 7 offers a direct performance comparison of the DroQ agent on the `Walker-v0` task at communication frequencies of 11Hz and 2Hz. Interestingly, the agent operating at 2Hz shows quicker and more stable convergence. We suspect that the lower communication frequency functions similarly to 'frame skip', a widely utilized technique in reinforcement learning. Frame skipping helps to reduce the number of actions an agent takes, thereby simplifying the decision-making processes. This method may explain the more efficient convergence observed with the 2Hz frequency.

### A.5 RunAway-v0 Strategies

Figure 8 illustrates the distinct strategies developed by the algorithms. Specifically, Figure 8b shows the final distance measured, while Figure 8a displays the mean action taken over the entire episode.

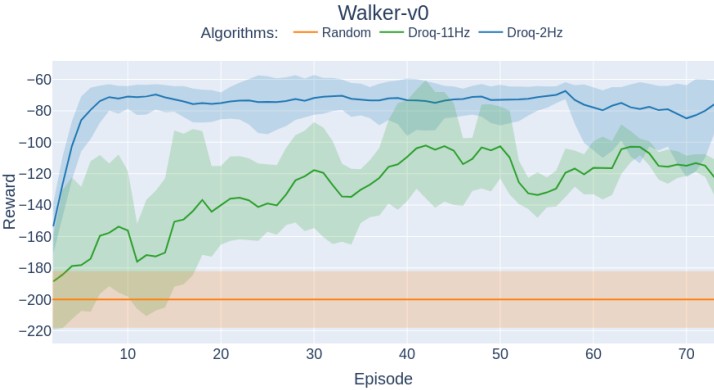

Figure 7: Comparison of communication frequencies for the DroQ agent on the `Walker-v0` task, illustrating the differences between the operational frequencies of 11Hz and 2Hz.

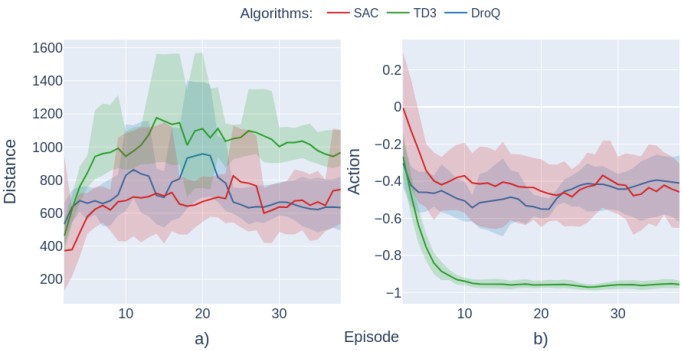

Figure 8: Final distance and (mean) action taken over one episode for the `RunAway-v0` task.

### A.6 Online Training Parameter

Table 12 displays the hyperparameter used in all our experiments.

### A.7 Dataset Generation Process

The expert dataset for each robot configuration was generated by training a Soft Actor-Critic (SAC) agent to solve the respective task and record transitions over 100 episodes on the real robot. The random dataset was created by executing a random policy for 100 episodes. For example, the collection process for the Walker robot took about an hour, yielding approximately 10,000 transitions. Details such as mean reward, number of transitions, and collection episodes for each dataset can be found in Table 11. The dataset is available on Hugging face.

### A.8 Offline Training Parameter

For the online algorithms, we used the same parameters in our offline rl experiments as in the online experiments.

We trained the models for various tasks and datasets with different update counts 13. The `RunAway-v0` task was trained for 2,000 updates on both the expert and random datasets. For the `Spinning-v0` task, we used 5,000 updates across both datasets. The `Walker-v0` task required 10,000 updates for both expert and random datasets. Finally, the `RoboArm-v0` task was trained for 10,000 updates on the random dataset and 5,000 updates on the expert dataset.

| Parameter | DroQ | SAC | TD3 |
|---|---|---|---|
| Learning Rate (lr) | $3 \times 10^{-4}$ | $3 \times 10^{-4}$ | $3 \times 10^{-4}$ |
| Batch Size | 256 | 256 | 256 |
| UTD Ratio | 20 | 1 | 1 |
| Prefill Episodes | 10 | 10 | 10 |
| Number of Cells | 256 | 256 | 256 |
| Gamma | 0.99 | 0.99 | 0.99 |
| Soft Update $\epsilon$ | 0.995 | 0.995 | 0.995 |
| Alpha Initial | 1 | 1 | - |
| Fixed Alpha | False | False | - |
| Normalization | LayerNorm | None | None |
| Dropout | 0.01 | 0.0 | 0.0 |
| Buffer Size | 1000000 | 1000000 | 1000000 |
| Exploration Noise | - | - | 0.1 |

Table 10: Hyperparameter for the agents DroQ, SAC, and TD3

| Task | Mean Reward | Expert Transitions | Random Transitions | Episodes |
|---|---|---|---|---|
| Walker-v0 | $-69.12$ | $9,244$ | $10,000$ | 100 |
| RoboArm-v0 | $-9.87$ | $1,297$ | $10,000$ | 100 |
| RunAway-v0 | $18.04$ | $1,987$ | $1,612$ | 100 |
| Spinning-v0 | $8981.19$ | $5,000$ | $5,000$ | 100 |

Table 11: Dataset Statistics

## A.9   Network Architecutre

Throughout the experiments, all algorithms utilize the same architecture for the policy, Q-functions, and value functions (where applicable). Each network is structured as a three-layer multilayer perceptron (MLP), with specific `TorchRL` actor modules used for the policy, depending on the algorithm. For more details, we refer readers to the code repository: GitHub.

The only variation in architecture occurs when incorporating pixel-based observations. In this case, a convolutional neural network (CNN) is used to encode the image data. These encodings are then concatenated with the sensor-based encodings, and the combined embeddings are passed through a shared MLP. Detailed implementation specifics can be found in the GitHub repository.

| Parameter | BC | IQL | CQL |
|---|---|---|---|
| Learning Rate (lr) | $3 \times 10^{-4}$ | $3 \times 10^{-4}$ | $3 \times 10^{-4}$ |
| Batch Size | 256 | 256 | 256 |
| Number of Cells | 256 | 256 | 256 |
| Gamma | - | 0.99 | 0.99 |
| Soft Update $\epsilon$ | - | 0.995 | 0.995 |
| Loss Function | L2 | L2 | L2 |
| Temperature | - | 1.0 | 1.0 |
| Expectile | - | 0.5 | - |
| Min Q Weight | - | - | 1.0 |
| Max Q Backup | - | - | False |
| Deterministic Backup | - | - | False |
| Num Random Actions | - | - | 10 |
| With Lagrange | - | - | True |
| Lagrange Threshold | - | - | 5.0 |
| Normalization | LayerNorm | None | None |
| Dropout | 0.01 | 0.0 | 0.0 |
| BC Steps | - | - | 1,000 |

Table 12: Hyperparameter for the agents BC, IQL, and CQL

| Task | Expert | Random |
|---|---|---|
| RunAway-v0 | 2,000 | 2,000 |
| Spinning-v0 | 5,000 | 5,000 |
| Walker-v0 | 10,000 | 10,000 |
| RoboArm-v0 | 5,000 | 10,000 |

Table 13: Number of offline training updates for each task and dataset

