# OpenReview forum: "BricksRL: A Platform for Democratizing Robotics and  Reinforcement Learning Research and Education with LEGO"
_NeurIPS.cc/2024/Conference — NeurIPS 2024 spotlight_

### Official Review · Reviewer_Tpu8 · 2024-07-11

**Soundness:** 3
**Presentation:** 2
**Contribution:** 3
**Rating:** 7
**Confidence:** 4

**Summary:**

The paper presents a framework for training reinforcement learning policies on LEGO robots in the real world.
BricksRL integrates the LEGO robotics hub PyBricks and a reinforcement learning library TorchRL, providing an easy interface to implement and deploy RL algorithms. A robust infrastrure for robot-environment-algorithm communication is implemented for real-world deployment.
A camera is integrated in the system beyond LEGO’s standard sensor set, which expands the platform’s capabilities.
Experiments on various tasks on three LEGO robots (2wheeler, walker, and robotarm) are demonstrated, validating the system’s capabilities for real-world robotics applications. Ablations are studied to compare sim2real transfer with learning from the real world, and w/ or wo/ camera sensor inputs.

**Strengths:**

Constructing robots with LEGO parts is low-cost and widely available. This paper demonstrates deploying reinforcement learning on LEGO robot, which is very inspiring for low-cost robot research.

LEGO parts are modular and reusable, allowing the user to tailor robot designs to specific tasks. This provides a nice platform for cross-embodiment policy research, task-specific robot design, etc.

The paper provides an effective solution to deploy reinforcement learning algorithms on low-cost robots, by integrating PyBricks and TorchRL, and implementing a robot-emvironment-algorithm communication via bluetooth connection. The system is proved to be robust by deploying real-world reinforcement learning policies.

Detailed experimental results on three LEGO robots are demonstrated, showcasing the system’s capability to train real-world rl policies on low-cost robots.

**Weaknesses:**

Integrating LEGO hub with BricksRL, due to communication overhead, the system frequency is only 11 Hz, which limits the capability of the system to perform certain tasks such as dynamic manipulation tasks.

It would be insightful for the authors to provide more discussions and ablations on the robustness of the system. The paper noted that the system has low communication speed, the lack of millimeter-precise constructions, backlash, and noisy sensors. But does the policy learn to be robust to these noises during training? Or are there certain strategies that authors employ to deal with these issues?

**Questions:**

Which simulator is used?

Is RoboArm-mixed-v0 trained only using real-world data? Perhaps because sim2real transfer with image is hard?

Is the trained policy robust to some slight deviations of robot hardware during explorations, such as shifted parts, sensor drifting, and system latency?

**Limitations:**

Yes, the authors addressed the limitations well.

---

> ### Author Rebuttal · Authors · 2024-08-02
>
> Response Weaknesses:
>
> Thank you for your assessment of the weaknesses, we are happy to address these points to provide more clarity.
>
> We recognize the potential benefits of increased communication speed and are willing to improve this. We've contacted PyBricks directly to collaborate on enhancing communication rates and they claim to have better protocols almost ready. Once implemented they will be made available in BricksRL immediately. However, we hardly had any problems due to the slower communication speed. As described in the paper, lower communication even led to more stable and faster learning for the walker robot.
>
> The policies learnt were robust notwithstanding existing backlashes and noise. For the real-world training, we did not take any measures to stabilize the learning as it was not necessary. However, as mentioned in the paper, for the sim2real transfer with the policies trained in simulations we added noise to the actions to simulate backlashes and sensor noises. This was sufficient to learn stable policies that could be evaluated successfully in the real world.
>
> Response Questions:
>
> - The robots were mainly trained online in the real world. We only developed two simulations to demonstrate the sim2real capabilities. Our approach to simulating LEGO robot models in BricksRL is straightforward but effective. In the two simulation environments, we simulate the transition dynamics by directly applying the actions to the simulated motor angles. This works well because our action space represents the delta angles for robot movement, allowing for a simple yet accurate simulation of the robot's behavior.
> - Yes, the RoboArm-mixed-v0 is fully trained in the real world, we did not develop any simulator for this environment. The fact that LEGO can be used in this setting proves the value of the LEGO platform for RL research.
> - Yes, the trained policies are robust to different sensor noises and backlashes. Different sensor noises/deviations are more harmful than others. For example, we did notice that during training of the RoboArm-mixed-v0 with image inputs lightning conditions are very important and need to be stable during the whole training. However, this is more specifically for the contour detection of the ball for reward calculation and not so much for policy learning.

---

> > ### Comment · Reviewer_Tpu8 · 2024-08-09
> >
> > Thank you for addressing my comments and questions. The idea and system setup of the paper are novel and inspiring, with sufficient experiments being demonstrated. I have raised my score.

---

### Official Review · Reviewer_tHkr · 2024-07-12

**Soundness:** 3
**Presentation:** 2
**Contribution:** 4
**Rating:** 6
**Confidence:** 5

**Summary:**

This paper presents BrickRL which is a system for using reinforcement learning within the context of lego robotics. The paper provides an overview of their setup, how they used TorchRL and PyBricks to interface with the the lego robots. They also provide results that show the feasibility of this system to use off-policy algorithms to train in the real world.

**Strengths:**

Originality: From my undersanding, this is the first paper to combine the lego robotics setup that has been popular in education for a while with RL. They cite other efforts to create low-cost robotics but even those are significantly more expensive than a simple lego setup, which further improves the originalitiy in my mind as a first of its kind very low-cost and flexible setup.

Quality: The main claim the authors make in this paper is their method is able to train agents on lego robots that perform well in the real world which they do. It would have been nice to mix both off and on policy algorithms in your results as well as maybe some offline RL just to clearly show the vercitility of your method. That being said, the claim of "agents can learn for lego RL" is clearly answered affermatively (at least for off-policy algorithms) from their results.

Significance: Major. This paper on the surface might seem simple as it is just combining lego with RL but in my opinion the significance of low-cost robotics for RL is massive. Researchers with less funding using this system for their work to test different algorithms on low-cost robots that are extremely easy to repair. The results are good enough that this system is clearly working and has the potential to address many accessibility issues with RL in robotics.

**Weaknesses:**

Clarity: The paper has no major errors but can be confusing as it does state things that feel like implementation details in the main paper that should likely go in the appendix and did make it harder to read.
Possible improvements:
- I think the PyBricksHubClass paragraph feels out of place and a little confusing. This could probably be just a simple "PyBricks provides a class that..." and simply state the benefits. I don't think the fact that it uses BLE is important.
- The BaseEnv paragraph also feels out of place. I think that putting it in there and saying that you can use a BaseEnv to create custom environments is redundant. The real benefit here is the second paragraph about enviornment transforms and other models.
- The useage of TensorDict feels just like an implementation detail, while it might be critical, these three paragraphs in 3.2 all feel like they could be one or two sentences and then you can move on and save more room for results.
- Simlar comment for Client Script and Robot-Environment Communication. It seems just like an implementation detail that might even be completely ok in the appendix. But Communication Speed is a very interesting section that I really like (addresses a possible limitation).

**Questions:**

Some of these questions are going to be in the unanswered limitations as well.
- How long do these robots last? Do the motors wear down during training? Are they expensive to replace?
- Could you use a wired connection for training? I realize it causes some issues but would it increase training Hz?
- How much did the setups you used cost?

**Limitations:**

The authors don't address many limitations in their paper which I do wish they more clearly did. Scattered around are a few limitations addressed like communication frequency but I think that limitations could be more clearly thought of and addressed.

---

> ### Author Rebuttal · Authors · 2024-08-02
>
> Response Weaknesses:
>
> We greatly appreciate your valuable input and suggestions for improving the clarity of our paper. We are pleased to incorporate the proposals as far as possible in the final version of our paper.
>
>
> Response Questions:
>
> We're happy to provide more details about our LEGO robot setups:
>
> Robot longevity and motor wear:
> - Battery life varies depending on the robot's complexity and usage intensity. For example, our walker and RoboArm models typically operate for 3-4 hours on a single charge, but it is possible to directly provide power via a USB connection to a larger battery pack.
> - Static robots like the RoboArm can be directly connected to a power supply via a USB for extended operation in training and execution
> - Throughout all our experiments and tests over a couple of years, we haven't experienced any part failure, indicating good durability. If replacement is needed, individual motors cost between 40-60€.
>
> Wired connections for training:
> - This is possible for stationary robots using the USB connection.
> - We recognize the potential benefits of increased communication speed. PyBricks is already experimenting with better communication channels and they will be made available in BricksRL immediately.
>
> Cost of setups:
> - Our setups utilize components that can be largely found in the LEGO Education kit and extension kit.  The current price range for these sets is approximately 700€.
>
>
>
> Response Limitations:
>
> The main limitations of LEGO components are in the tolerance of the gears, precisions of the encoders, and torque of the motors. However, in our experiments, we have shown that BricksRL is capable of working around those. Our extensive testing and experiments with the LEGO robots have yielded very robust results.
>
> We further show the robustness in the sim2real experiments, where policies trained in idealized simulations were directly transferable to real robots with minimal issues. Our approach of adding small amounts of noise to simulated observations to mimic backlash and sensor inaccuracies proved effective in bridging the reality gap.
>
> Regarding the limitation of communication speed, as mentioned above Pybricks is working on better communication channels, and once available those will be immediately included in BricksRL.

---

> > ### Comment · Reviewer_tHkr · 2024-08-08
> >
> > Thank you for answering my questions and comments. I think your work provides novelty and impact to a broad range of researchers without access to expensive equipment. I'm going to leave my rating unchanged.

---

### Official Review · Reviewer_R3sL · 2024-07-12

**Soundness:** 3
**Presentation:** 2
**Contribution:** 3
**Rating:** 5
**Confidence:** 3

**Summary:**

This work proposes a flexible and cost-effective platform named BricksRL with LEGO builds aiming to lower the cost of RL research and education. Experiments also demonstrated that LEGO robots can be trained within 120 minutes on normal computers to achieve simple tasks such as moving, walking, and grasping. An offline dataset is also provided for offline RL or imitation learning.

**Strengths:**

- Practical utility has been demonstrated by showing the deployment results of RL methods on three LEGO robots such as SAC, TD3, and DroQ.
- This LEGO robot platform is low-cost and flexible with clear instructions and open-source code.

**Weaknesses:**

- BricksRL’s value as an educational platform is clear but the scientific contribution is limited because there's no innovation proposed on algorithms.
- A dataset constructed with BricksRL may be of greater value for offline reinforcement learning and imitation learning.
- The tasks that BricksRL supports are relatively simple, such as walking, spinning, and reaching.

**Questions:**

- In what way can LEGO robot models be converted to models that can be used in simulation?
- Does BricksRL also support urdf model construction or the integration into mainstream simulators like Mujuco or Isaac Sim/Gym?

**Limitations:**

As mentioned in Weaknesses, the scientific contribution of this work is limited.

---

> ### Author Rebuttal · Authors · 2024-08-02
>
> Response Weaknesses:
>
> Thank you for your thoughtful feedback, we appreciate your recognition of its practical utility and educational value. However, we respectfully disagree with the assessment that the scientific contribution is limited.
>
> We would like to emphasize several key points that demonstrate the significant scientific value of BricksRL:
> - BricksRL is the first platform to seamlessly integrate state-of-the-art reinforcement learning algorithms with hardware that is economical, modular, mass-produced, and easy to use like LEGO. This allows everybody in the world to focus on RL algorithms but test their ideas on real robots straightaway. There are over 1M LEGO hubs sold worldwide which are compatible with Pybricks and BricksRL as of today. We cannot emphasize enough how transformative this aspect is in the field of AI for robotics.
> - By enabling the creation of low-cost, customizable robots, BricksRL addresses a critical challenge in RL research - the ability to conduct reproducible experiments on multiple real robots at once. This scalability and reproducibility is crucial for advancing RL algorithms in real-world settings.
> - We have shown with BricksRL that complex RL algorithms like SAC, TD3, and DroQ can be successfully applied to train LEGO robots end-to-end in the real world within reasonable time frames. This is a necessary validation and demonstration of the platform towards making real-world RL more accessible and practical.
> - We have also demonstrated the ability to incorporate non-LEGO sensors to increase its potential for broader applications in robotics research, beyond just LEGO components. LEGO itself might provide more sensors in the future.
> - By providing built plans and training results, BricksRL contributes to the reproducibility of RL experiments in robotics, a crucial aspect of scientific research often challenging in real-world settings. BricksRL has clear educational benefits, but we have also created it for its research potential. We believe that BricksRL's contribution represents a significant step towards democratizing real-world RL research in robotics, addressing key challenges in the field such as cost, scalability, and the reality gap.
>
> Offline datasets:
>
> We have already generated valuable offline datasets with our LEGO robots. As part of our revised paper submission, we are making these datasets publicly available in the code repository to facilitate research in offline RL that is directly available in TorchRL.
>
> We have curated datasets for the three robot configurations: 2Wheeler, Walker, and RoboArm. These datasets encompass both expert and random data, specifically tailored to the tasks explored in our experiments.
>
> For the 2Wheeler, we offer datasets for the 'RunAway-v0' and 'Spinning-v0' environment tasks. The Walker robot dataset focuses on the 'Walker-v0' task, while the RoboArm robot dataset addresses the 'RoboArm-v0' task.
> The expert data for each dataset was carefully collected using the following process: We first trained a Soft Actor-Critic (SAC) agent to successfully complete the specific task. Upon achieving satisfactory performance, we evaluated the agent over 100 episodes per task, recording all transitions. For example, the evaluation process took approximately one hour for the Walker robot and yielded 10,000 expert transitions.
>
> To complement the expert data, we also generated random datasets. These were created by executing a random policy in the environment, similarly for 100 episodes per task. This process resulted in eight distinct datasets:
> 1. RunAway-v0-expert
> 2. RunAway-v0-random
> 3. Spinning-v0-expert
> 4. Spinning-v0-random
> 5. Walker-v0-expert
> 6. Walker-v0-random
> 7. RoboArm-v0-expert
> 8. RoboArm-v0-random
>
> We trained successfully offline-RL algorithms, including BC, TD3+BC, IQL, and CQL using these datasets. This allows researchers to compare the performance of various RL algorithms, both offline and online, when trained on expert demonstrations versus random interactions. Such comparisons can provide valuable insights into the effectiveness and adaptability of different RL methods across various data quality levels and robot configurations.
>
>
> Response Questions:
>
> Currently, our approach to simulating LEGO robot models in BricksRL is straightforward but effective. In our experiments, we simulated the transition dynamics by directly applying the actions to the simulated motor angles. This works well because our action space represents the delta angles for robot movement, allowing for a simple yet accurate simulation of the robot's behavior.
> BricksRL does not yet support URDF model construction, which would allow for more sophisticated simulations and easier integration with mainstream simulators like MuJoCo or Isaac Sim. However, a few LEGO CAD (https://www.leocad.org/) are actually available. One could build completely virtual robots using only LEGO parts. The complete list of parts is then listed. The LEGO ecosystem is maybe unsurprisingly very rich. It would be therefore time-consuming but relatively straightforward to build simulated LEGO robotics. We are considering this avenue of developing MuJoCo-based LEGO simulations for the future.  This is, however, secondary, to prove that RL research for robotics can be done at all with LEGO which is the outcome of this work.

---

> > ### Comment · Reviewer_R3sL · 2024-08-10
> >
> > Thank you for answering my previous doubts about scientific contributions and simulation integration. The offline dataset you additionally provide will be of great value to the public for basic RL research. The real-world experiment is sufficient to prove the scalability and reproducibility of LEGO robots. I have re-evaluated this work and will raise my score accordingly. However, I still hope this work can be extended to support more complex manipulation tasks that integrate interactions with objects to bring broader impact.

---

### Official Review · Reviewer_6ArU · 2024-07-13

**Soundness:** 3
**Presentation:** 3
**Contribution:** 3
**Rating:** 6
**Confidence:** 4

**Summary:**

This paper demonstrates that cheap LEGO robots can be trained in under 120 mins on a laptop to perform simple tasks using sim2real approaches. The paper is motivated by making robot learning research more accessible for educational settings. A software framework called BricksRL is released in open source that integrates PyBricks and TorchRL with Gym environments for controlling modular LEGO-based components, motors and sensors.

**Strengths:**

* The paper is clearly written, with a strong motivation section and a sound implementation of RL algorithms.
* The paper is accompanied with instructions on how to build LEGO robots paired with BrickRL for exploring RL/control/robot learning algorithms much more cheaply than with industrial robots.
* The finding that simple tasks can be trained for in a couple of hours on a normal laptop is compelling for educational experimentation.
* LEGO robots are easily extensible, so one may envision quickly designing higher-degree-of-freedom robots for research

**Weaknesses:**

* The core contributions of this paper are not technically novel - there are no new algorithms introduced, and BricksRL is a standard Gym framework for RL training.
* The tasks demonstrated are relatively simple (walking, spinning, reaching).  Hence, it is not clear how much value this framework provides beyond educational experimentation in simulation.
* The RL algorithms benchmarked may not reflect current state of the art approaches - but this is not the prime focus of this paper. The variance in the plots seems to be extremely high.

**Questions:**

- The paper is currently lacking details on policy architectures used for training. Please add.
- Are any of the policies vision-based?  As the number of parameters increases, the training time will also increase. This will stress the reliability of LEGO hardware - possibly breakages that may complicate learning on more complex tasks. Some expanded discussion on this would be helpful.

**Limitations:**

Yes, to some extent. One issue with low-cost robots is easy wear and tear precluding large-scale data collection. The presence of backlash, noisy sensing may also limit experimentation for more precise tasks. Please expand more on limitations in this regard.

---

> ### Author Rebuttal · Authors · 2024-08-02
>
> Response Weaknesses:
>
> - Our primary contribution lies in providing accessible, low-cost hardware integration for real-world robotics experimentation. Our goal is to lower the barrier to entry for robotics research and education, enabling a wider audience to engage with physical robots beyond simulations. LEGO parts are widely available, robust, and reproducible because they are mass-produced. We are not aware of any other robotic platform with such characteristics, making this paper a first instance of that.
>
> - The demonstrated tasks serve as proof-of-concept examples to validate the framework's functionality. It is not at all obvious that LEGO can be used for RL tasks, as it is not designed for that. Gears have generous tolerance, the stepping motors offer encoders but the precision is about 1 degree, and structurally, they are made of bricks. In this paper, we show that all this hardware can be used effectively for RL.
>
> - Our focus is on demonstrating the framework's compatibility with a range of algorithms rather than pushing the boundaries of RL performance given that this is a totally new platform. The outcome is a validated platform for the democratization of robotics. However, we made sure to implement the RL framework using the official RL library from PyTorch. This allows direct usage of all state-of-the-art RL algorithms. Since submission, we have already expanded our example experiments to include several offline RL algorithms, such as BC, TD3+BC, IQL, and CQL, trained on collected datasets. We've also integrated pre-trained foundation models like VIP transform from TorchRL. Which can be used to give reward signals to sparse reward tasks but can also be utilized to label hand-collected datasets that have no reward signal. VIP can enable simple, few-shot offline RL on real-world robotic tasks for low data quantities. We are making available these examples and the offline datasets with the revised version of the paper.
>
> - The high variance in the plots is potentially stemming from the noisy nature of real-world robotics. This variability itself is an interesting area for future research, investigating robust learning methods in noisy, real-world environments.
>
> Response Questions:
>
> We apologize for the lack of details on policy architectures in the current version of the paper. We will add this information in the appendix. All examples are available in full detail in the GitHub repository. Our default architecture for both policies and Q-functions is a multilayer perceptron (MLP) with three linear layers and ReLU activation functions. We also incorporate dropout and layer normalization specifically for the DroQ algorithm.
>
> - For the vision-based environment (roboarm-mixed-v0), we employ a hybrid architecture. This combines image observations with state observations of the robot angles. Specifically, image observations are processed through a convolutional neural network (CNN), while state observations are encoded using an MLP. The resulting image and state embeddings are concatenated and passed through a final output head. We have added a detailed description of this architecture in the appendix of the paper.
>
> - Regarding the concern about increased training time and potential stress on LEGO hardware. We have run the same LEGO components for almost two years to prepare this paper with zero faulty parts. LEGO parts are created for kids and mass-produced so they don't break.
>
> Response Limitations:
>
> Thank you for raising these important points about the limitations of low-cost robots. The main limitations of LEGO components are in the tolerance of the gears, precisions of the encoders, and torque of the motors. However, in our experiments, we have shown that BricksRL is capable of working around those. Our extensive testing and experiments with the LEGO robots have yielded very robust results. Throughout our trials, we did not encounter a single failure of any motor or gear, which speaks to the reliability of these components even under repeated use.
> Regarding backlash and sensor noise, these factors are certainly present in our setup. However, their impact on policy learning and execution has not been significant. This is evidenced by our successful sim2real experiments, where policies trained in idealized simulations were directly transferable to real robots with minimal issues. Our approach of adding small amounts of noise to simulated observations to mimic backlash and sensor inaccuracies proved effective in bridging the reality gap.
> The architecture ensures that the computational load of training and inference remains separate from the physical robot, mitigating concerns about hardware reliability or potential breakages due to increased model complexity. The LEGO components are primarily involved in the physical execution of actions and collection of observations, tasks which are not affected by increases in model size. We have added this information to the main manuscript.

---

### Author Rebuttal · Authors · 2024-08-07

We have added this subsection (PDF) to the paper.

---

### Decision · Program_Chairs · 2024-09-25

**Decision:**

Accept (spotlight)

**Comment:**

The reviewers and the AC were impressed by the idea and system setup (software + hardware). Thanks for making the data available. The system has been extensively tested. The rebuttal managed to alleviate all major doubts of the reviewers.I believe this is a very interesting and promising piece of work that provides a low-cost benchmark for real-robot RL experiments with a large potential for customization and extensions.